# MATCH PREDICTION FROM GROUP COMPARISON DATA USING NEURAL NETWORKS

## ABSTRACT

We explore the match prediction problem where one seeks to estimate the likelihood of a group of $M$ items preferred over another, based on partial group comparison data. Challenges arise in practice. As existing state-of-the-art algorithms are tailored to certain statistical models, we have different best algorithms across distinct scenarios. Worse yet, we have no prior knowledge on the underlying model for a given scenario. These call for a unified approach that can be universally applied to a wide range of scenarios and achieve consistently high performances. To this end, we incorporate deep learning architectures so as to reflect the key structural features that most state-of-the-art algorithms, some of which are optimal in certain settings, share in common. This enables us to infer hidden models underlying a given dataset, which govern in-group interactions and statistical patterns of comparisons, and hence to devise the best algorithm tailored to the dataset at hand. Through extensive experiments on synthetic and real-world datasets, we evaluate our framework in comparison to state-of-the-art algorithms. It turns out that our framework consistently leads to the best performance across all datasets in terms of cross entropy loss and prediction accuracy, while the state-of-the-art algorithms suffer from inconsistent performances across different datasets. Furthermore, we show that it can be easily extended to attain satisfactory performances in rank aggregation tasks, suggesting that it can be adaptable for other tasks as well.

## 1 INTRODUCTION

The most elementary form of comparisons is pairwise: we often compare a pair of items and make judgments as to which one is of higher utility or simply preferable over the other. With a large amount of such comparison data, one can consider various interesting tasks. One may wish to predict future outcomes of unseen matches, and also to rank alternatives in order of utility or preference.

Challenges arise in carrying out these tasks. Almost all existing state-of-the-art algorithms have been developed under the assumption that given a scenario, there exists a certain underlying model which governs statistical patterns of comparison data (see Section 2 for details). As such, we have different best-performing algorithms across distinct scenarios. This traditional approach, which begins by assuming certain models to develop algorithms, comes with limitations in practice.

First, it gives rise to *inconsistent* performances. No single algorithm can perform consistently well in a wide range of scenarios, since it has been tailored to a specific model. Second, it is hard to know the underlying model without expert domain knowledge. In its absence, we have little choice but to find an appropriate algorithm via trial-and-error. Third, the model can be inherently complex for any existing algorithm to be effective. Sometimes groups of items are compared, thus the effects of interactions among in-group items come into play, further complicating the model.

In this work, we propose a unified algorithmic framework aimed to overcome these barriers. We focus on the match prediction problem where one wishes to estimate the likelihood of a group of $M$ items preferred over another, based on partially observed group comparison data among a collection of $n$ items. One can imagine that such group comparison data may bear complex statistical patterns due to a combination of two underlying models: the *interaction* model which governs the effects of in-group interactions in determining the utility or preference of a group; and the *comparison* model which governs the statistical patterns of pairwise group comparison data. Hence, designing a novel framework hinges heavily upon accurate inference of these underlying models.

**Main contribution.** We incorporate deep learning techniques into our framework design. This enables us to infer the underlying models from real-world data obtained from a given application, and thus to achieve consistently high performances on a variety of datasets from diverse real-world applications where match prediction tasks are of interest.

To this end, we build on progress made through analysis in well-defined statistical models. We gain insights instrumental to the progress by looking into existing state-of-the-art algorithms in related and long-studied tasks such as rank aggregation (Negahban et al., 2016; Hunter, 2004; Huang et al., 2006; 2008). We find that most of them share a key element. They all exhibit so-called *reward-and-penalty* mechanisms in estimating the utilities of individual items.

To be more specific, they reward an item more greatly for winning (or being more preferred) in a disadvantageous comparison where its group is weaker than the counterpart. Likewise, they penalize it more greatly for losing (or being less preferred) in an advantageous one. In addition, the magnitudes of rewards and penalties are proportional to the contribution of the individual item to its group.

This structural similarity across the state-of-the-art algorithms has attracted our attention. Through some manipulation, we find that they all employ the same basic rule for estimating individual utilities (see (6) in Section 4 for details). The terms corresponding to rewards and penalties turn out to vary as either one of the two underlying models changes. This observation has inspired us to incorporate neural networks into our framework design.

The novelty of our design is salient in an ablation study where we compare it with a simple design. As an initial effort, a single-layer neural network has been employed to predict winning probabilities of unseen group matches (Menke & Martinez, 2008). It has shown a promising result, demonstrating prediction accuracy to be improved on a real-world online game dataset, but also exhibited a scalability issue. It requires one input node per item, making it prohibitive to be extended to real-world applications with a large number of items. Leveraging more advanced architectures (see Figures 1 and 2) motivated by observant analysis as emphasized, our design not only addresses such a scalability issue by design, but also outperforms the single-layer neural network. The merits of our design are evaluated against the single-layer neural network and other state-of-the-art algorithms through extensive experiments on a variety of synthetic and real-world datasets (see Section 5).

Using synthetic datasets, we demonstrate that our approach can achieve the performances of the state-of-the-art algorithms in the models for which they have been specifically developed. We investigate four models. Three consider various extensions of the Bradley-Terry-Luce model (Bradley & Terry, 1952) to the group comparison scenario. The other is a generalized version of the Thurstone model (Herbrich et al., 2007) widely used in skill rating systems of online games. As a result, we show that our framework consistently yields the best performances across all of these datasets (near-best in some cases), while the other state-of-the-art algorithms suffer from inconsistent performances across different models.

Using real-world datasets, we also demonstrate that our framework performs consistently well across diverse real-world applications. We investigate five real-world datasets (sources in Footnote 6). One is a crowd-sourced image classification dataset, another is a collection of movie ratings, and the other three are match records from online games. We consider, in addition to the cross entropy loss, the prediction accuracy as another metric (defined in (9)). As a result, we show that our framework consistently yields almost the best performances across all of these datasets in terms of both metrics.

We also show that our framework can be easily extended to achieve the best performance in rank aggregation tasks where one seeks to rank items in order of utility or preference. Using a real-world dataset of movie ratings, we demonstrate that our framework yields the best performances in terms of two well-known metrics (see Footnote 10): Kendall tau distance (Kendall, 1938) and normalized discounted cumulative gain (Järvelin & Kekäläinen, 2002). This result suggests that it can potentially be adaptable for other tasks as well.

## 2   RELATED WORK

The most related prior works are (Huang et al., 2006; 2008; Li et al., 2018; Herbrich et al., 2007) where plausible statistical models (some long-established and some widely used in practice) for in-

group interactions and group comparisons are assumed, and statistical analysis is carried out to a great extent. We use the algorithms developed in these models and their variants for main baselines.

The problem of estimating individual utilities from group comparison data has been investigated in (Huang et al., 2006; 2008). They considered extensions of the BTL model where the group utility is either the sum or the product of individual utilities, and group comparison data follow the BTL model in terms of two group utilities (which we call the BTL-sum and BTL-product models respectively).

A more advanced in-group interaction model has been explored in (Li et al., 2018). They considered a scenario where a pair of individuals in a group leads to a synergy, which contributes to the group. The group utility is represented as the sum of two quantities (the HOI model): (1) the sum of individual utilities and (2) the sum of the products of all pairs of individual utilities. A general scenario, where any $k$-tuple of individuals in a group leads to a synergy, has been considered in (DeLong et al., 2011).

In (Herbrich et al., 2007), they assumed individual utilities to be centered around a mean following a Gaussian distribution and viewed the group utility as their sum (the Thurstone model). The algorithm therein is widely used in skill rating systems of online games where groups of users compete.

Employing neural networks has been considered as an initial effort to predict winning probabilities of unseen group matches (Menke & Martinez, 2008). It has been shown that a single-layer neural network can fit some variants of the BTL model (Huang et al., 2008) and improve prediction accuracy through experiments on a real-world online game dataset.

Some other works have also employed neural networks to exploit hidden models, for example, in information retrieval (Burges et al., 2005; Richardson et al., 2006; Guo et al., 2016) and community detection (Chen et al., 2018b) in graph domains (Scarselli et al., 2009; Schlichtkrull et al., 2018).

## 3 PROBLEM SETUP

We investigate the match prediction problem where we seek to predict comparison outcomes for unobserved pairs of groups given a collection of pairwise group comparison data. We consider the setting where each group consists of $M$ individual items.

We are given comparison observations between two groups of size $M$. To be more specific, each comparison consists of $(\mathcal{A}, \mathcal{B}, y_{\mathcal{AB}})$. $\mathcal{A}$ and $\mathcal{B}$ are groups of $M$ individuals, respectively. $y_{\mathcal{AB}}$ indicates which group wins (or is preferred):

$$y_{\mathcal{AB}} = \begin{cases} 1 & \text{if } \mathcal{A} \succ \mathcal{B}; \\ 0 & \text{otherwise,} \end{cases} \tag{1}$$

where $\mathcal{A} \succ \mathcal{B}$ indicates that group $\mathcal{A}$ wins over group $\mathcal{B}$. We denote by $\mathcal{D}_{\text{obs}}$ the set of observed comparisons, so all information we are given is $\{(\mathcal{A}, \mathcal{B}, y_{\mathcal{AB}})\}_{(\mathcal{A}, \mathcal{B}) \in \mathcal{D}_{\text{obs}}}$. The unobserved set of comparisons, which we wish to predict, is denoted by $\mathcal{D}_{\text{unobs}}$.

We consider the cross entropy loss as our metric, as it serves to quantify the discrepancy between two probability distributions. We define $\hat{y}_{\mathcal{AB}}$ as the estimate of $\Pr[y_{\mathcal{AB}} = 1]$. Our goal is to minimize

$$\frac{-1}{|\mathcal{D}_{\text{obs}}|} \sum_{(\mathcal{A}, \mathcal{B}) \in \mathcal{D}_{\text{obs}}} y_{\mathcal{AB}} \log \hat{y}_{\mathcal{AB}} + (1 - y_{\mathcal{AB}}) \log(1 - \hat{y}_{\mathcal{AB}}). \tag{2}$$

We develop our algorithm primarily based on the cross entropy loss, but for evaluation purposes, we also consider other metrics, such as prediction accuracy, the Kendall tau distance, and normalized discounted cumulative gain (see Footnote 10).

**Notation.** We denote by $[n] = \{1, 2, \ldots, n\}$ the set of all individual items. We use lowercase letters such as $i$ and $j$ to represent individual items, and caligraphic letters such as $\mathcal{A}$ and $\mathcal{B}$ to represent sets of items. We denote by $\{w_i\}_{i \in \mathcal{A}}$ the set of $w_i$'s for all $i \in \mathcal{A}$. We denote by $[w_i]_{i \in \mathcal{A}}$ a vector of $w_i$'s for all $i \in \mathcal{A}$ and its ordering information is provided in the context. We denote by $\hat{y}$ an estimate of $y$. We use subscripts as in $y_{\mathcal{AB}}$ when $y$ concerns a comparison between $\mathcal{A}$ and $\mathcal{B}$. We use superscripts as in $w^{(t)}$ when $w$ is updated iteratively. We use boldsymbols as in $\boldsymbol{w}$ to represent the set of $w_i$'s for all $i \in [n]$.

# 4 PROPOSED ALGORITHM

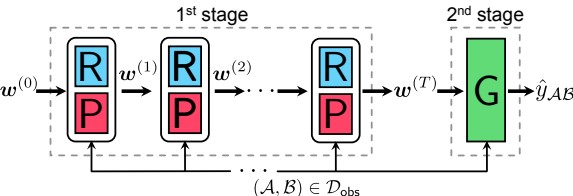

Figure 1: Overall architecture of our proposed algorithmic framework.

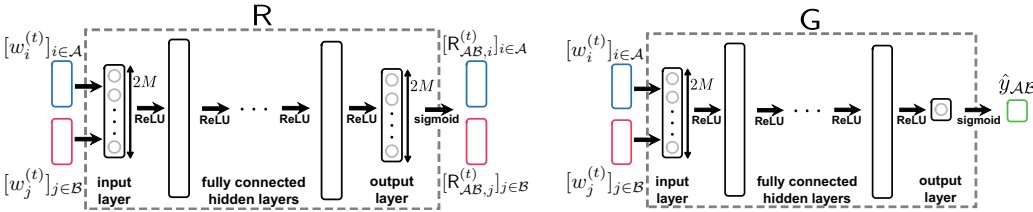

Figure 2: Detailed architectures of the key modules: module R (or P) (left) and module G (right).

We propose a neural network architecture that learns from observed group comparison data to predict unobserved group comparison outcomes. As presented in Figure 1, it consists of three modules, which we denote by R, P and G. Figure 2 presents the detailed architecture of R and P (left) which have the same structure but are separate modules with different weights, and that of G (right).

## 4.1 MOTIVATION

Our decision to incorporate two modules R and P into our architecture has been inspired by state-of-the-art algorithms that have been shown optimal (either achieving the minimal sample complexity or the global minima of cross entropy loss; details presented soon) under extensions of the well-established BTL model. Our main contribution lies in this design choice. Examining the algorithms in detail, we discover that they all share a similar mechanism in estimating individual utilities:

$(a)$ They all exhibit "reward" and "penalty" terms in the estimation process (details in (3) and (4)). They update an individual's utility estimate by rewarding the individual for contributing to its group's winning and likewise penalizing it for contributing to its group's losing.

$(b)$ These reward and penalty terms *vary* in form as the underlying models change.

$(c)$ The magnitudes of rewards and penalties depend on the power dynamics between groups. A greater reward is given to an individual when its group is relatively weaker compared to the opponent group, and likewise a greater penalty is given when its group is relatively stronger.

$(d)$ Their magnitudes also depend on the portion of an individual's contribution within its group. Suppose an individual's group wins (or loses) in a group comparison. The individual is given a greater reward (or penalty) when its contribution among others in the group is relatively greater.

Our algorithm design principles are centered around these key observations. We introduce two separate modules to represent rewards and penalties respectively. We employ deep neural networks into the modules so that they can serve to infer underlying models, which are unknown in practice.

**Reward and Penalty in State-of-the-art Algorithms.** We present details of the state-of-the-art algorithms developed under extensions of the BTL model. They illustrate all of the key observations.

- To begin with a simple case, *Rank Centrality* (Negahban et al., 2016) has been developed under the well-known BTL model where individual items are compared in pairs. It has been shown to achieve the minimal sample complexity for top-$K$ rank aggregation, whose task is to estimate the set of top-$K$ items, in certain regimes (Jang et al., 2017; Chen et al.,

2017). As in (1), we define $y_{ij}$ as 1 if $i \succ j$ and 0 otherwise, given a pair of individual items $i$ and $j$. Then, its individual utility update rule is[1]:

$$w_i^{(t+1)} \leftarrow w_i^{(t)} + \alpha \sum_{j:(i,j)\in\mathcal{D}_{\text{obs}}} \left( y_{ij} w_j^{(t)} - (1 - y_{ij}) w_i^{(t)} \right). \tag{3}$$

For item $i$, one can consider $w_j^{(t)}$ (next to $y_{ij}$) as the reward because it increases $w_i^{(t+1)}$ when $i \succ j$ ($y_{ij} = 1$), and $w_i^{(t)}$ (next to $(1-y_{ij})$) as the penalty because it decreases $w_i^{(t+1)}$ when $i \prec j$ ($y_{ij} = 0$). One can consider $\alpha$ as a step size in the update. Note that the reward is large when the opponent's utility estimate is large, since it can be considered that it has won in a tough match. Likewise, the penalty is large when its own utility estimate is large, since it has lost in an easy match. We can see the reward and penalty mechanisms, and the magnitudes of their influences based on the dynamics between groups. These correspond to observations $(a)$ and $(c)$.

- Majorization-Minimization (MM) for the BTL-sum model has been developed in (Hunter, 2004; Huang et al., 2006). We define $w_{\mathcal{A}}^{(t)} := \sum_{i\in\mathcal{A}} w_i^{(t)}$. Then, its individual utility update rule is[2]:

$$w_i^{(t+1)} \leftarrow w_i^{(t)} + \alpha_i \sum_{(\mathcal{A},\mathcal{B})\in\mathcal{D}_{\text{obs}}, i\in\mathcal{A}} \left( y_{\mathcal{A}\mathcal{B}} \cdot \mathsf{R}_{\mathcal{A}\mathcal{B},i}^{(t)} - (1 - y_{\mathcal{A}\mathcal{B}}) \cdot \mathsf{P}_{\mathcal{A}\mathcal{B},i}^{(t)} \right) \tag{4}$$

where

$$\mathsf{R}_{\mathcal{A}\mathcal{B},i}^{(t)} = \frac{w_{\mathcal{B}}^{(t)}}{w_{\mathcal{A}}^{(t)} + w_{\mathcal{B}}^{(t)}} \cdot \frac{w_i^{(t)}}{w_{\mathcal{A}}^{(t)}}, \quad \mathsf{P}_{\mathcal{A}\mathcal{B},i}^{(t)} = \frac{w_{\mathcal{A}}^{(t)}}{w_{\mathcal{A}}^{(t)} + w_{\mathcal{B}}^{(t)}} \cdot \frac{w_i^{(t)}}{w_{\mathcal{A}}^{(t)}}. \tag{5}$$

Note that the update rule (4) is similar to (3) of *Rank Centrality* in form, but the reward and penalty terms (5) are different. The interpretation is similar. The reward for item $i$ is large when the opponent group's utility estimate is large ($w_{\mathcal{B}}^{(t)}$). Note another factor: the larger the contribution of item $i$ within its own group ($w_i^{(t)}/w_{\mathcal{A}}^{(t)}$), the greater the reward. The same holds for the penalty. In addition to observations $(a)$ and $(c)$, we can also see the varying reward and penalty terms due to a different underlying model, and the effect of an individual's contribution within its group for rewards and penalties. These correspond to observations $(b)$ and $(d)$.

- MM for the BTL-product model has been developed in (Huang et al., 2008) and shown to achieve global minima in terms of cross entropy loss. Its individual utility update rule is described as in (4) but the reward and penalty terms are different[3]. Here, a similar interpretation applies again, and we can also see observations $(a)$–$(d)$ at play.

Our decision to incorporate another module G is as follows. To perform the match prediction task, we need not only individual utility estimates, but also in-group interaction and group comparison models that use such estimates to determine winning probabilities for pairs of groups. The role of G is to fit these underlying models during training. It takes as input the individual utility estimates from a pair of groups, which R and P help quantify, and predicts as output the probability of one group preferred over the other. The three modules interact as a whole to perform the task.

## 4.2 MODULES R AND P

The input and output of modules R (or P) are of dimension $2M$ (see Figure 2). $M$ can be an arbitrary integer greater than unity ($M > 1$). We set its value according to the dataset at hand.

---

[1]As in (Negahban et al., 2016), $\alpha = 1/d_{\max}$ where $d_{\max} := \max_i d_i$ and $d_i$ is the number of distinct items to which item $i$ is compared. Also, we describe *Rank Centrality* as an iterative algorithm (one way to obtain the stationary distribution of the empirical pairwise preference matrix) in order to highlight its inherent reward-and-penalty mechanisms.

[2]$\alpha_i = \left( \sum_{(\mathcal{A},\mathcal{B})\in\mathcal{D}_{\text{obs}}, i\in\mathcal{A}} \left( w_{\mathcal{A}}^{(t)} + w_{\mathcal{B}}^{(t)} \right)^{-1} \right)^{-1}$.

[3]$w_{\mathcal{A}}^{(t)} := \prod_{i\in\mathcal{A}} w_i^{(t)}, \alpha_i = \left( \sum \frac{w_{\mathcal{A}}^{(t)}}{w_{\mathcal{A}}^{(t)}+w_{\mathcal{B}}^{(t)}} \right)^{-1}, \mathsf{R}_{\mathcal{A}\mathcal{B},i}^{(t)} = \frac{w_{\mathcal{B}}^{(t)}}{w_{\mathcal{A}}^{(t)}+w_{\mathcal{B}}^{(t)}} \cdot w_i^{(t)}$ and $\mathsf{P}_{\mathcal{A}\mathcal{B},i}^{(t)} = \frac{w_{\mathcal{A}}^{(t)}}{w_{\mathcal{A}}^{(t)}+w_{\mathcal{B}}^{(t)}} \cdot w_i^{(t)}$.

They take the current utility estimates of the individuals in a given group comparison $(\mathcal{A}, \mathcal{B})$ as input, and produce the current R (or P) estimates for the individuals as output[4]. Note that the input and output dimensions are *independent* of the total number of items. Hence, our framework does not suffer from scalability issues in contrast to prior approach (Menke & Martinez, 2008) also based on employing neural networks. To make our algorithm robust against arbitrary orderings of the items within a group, we apply data augmentation. Given a sample, we create extra samples which represent the same outcome but have different item orderings. For example, given a sample $(\mathcal{A} = (1, 2), \mathcal{B} = (3, 4), y_{\mathcal{A}\mathcal{B}} = 1)$, we create an extra sample such as $(\mathcal{A}' = (2, 1), \mathcal{B}' = (4, 3), y_{\mathcal{A}'\mathcal{B}'} = 1)$. We also make our algorithm robust against arbitrary orderings of the two groups. That is, we create extra samples by changing the order of two sets $\mathcal{A}$ and $\mathcal{B}$ as well as $\mathcal{A}'$ and $\mathcal{B}'$. This technique helps train modules R and P in such a way that they become robust against two kinds of arbitrary orderings: item orderings within a group and group orderings.

All layers are fully connected. The activation functions between two layers are rectified linear units. The final activation function is the sigmoid function whose output ranges between 0 and 1.

Starting with an initial vector $\boldsymbol{w}^{(0)} \in \mathbb{R}^n$ (reflecting the current utility estimates), we finally obtain $\boldsymbol{w}^{(T)} \in \mathbb{R}^n$ by applying R and P repeatedly $T$ times. Each iteration is described as follows:

$$w_i^{(t+1)} \leftarrow w_i^{(t)} + \alpha \sum_{(\mathcal{A}, \mathcal{B}) \in \mathcal{D}_{\text{obs}}, i \in \mathcal{A}} \left( y_{\mathcal{A}\mathcal{B}} \cdot \mathsf{R}_{\mathcal{A}\mathcal{B}, i}^{(t)} - (1 - y_{\mathcal{A}\mathcal{B}}) \cdot \mathsf{P}_{\mathcal{A}\mathcal{B}, i}^{(t)} \right), \quad (6)$$

where[5] $\alpha = c / \max_i d_i$ and $d_i = |\{(\mathcal{A}, \mathcal{B}) : i \in \mathcal{A} \cup \mathcal{B}\}|$.

At the end of each iteration, we transform $\boldsymbol{w}^{(t+1)}$ to be zero-mean as in $w_i^{(t+1)} \leftarrow w_i^{(t+1)} - \sum_{i=1}^n w_i^{(t+1)}/n$, and unity-norm as in $w_i^{(t+1)} \leftarrow w_i^{(t+1)}/\|\boldsymbol{w}^{(t+1)}\|_2$. At iteration $t$, given $\boldsymbol{w}^{(t)}$, R and P produce positive real values in $[0, 1]$:

$$\mathsf{R}_{\mathcal{A}\mathcal{B}, i}^{(t)} = \mathsf{R}\left( i, [w_j^{(t)}]_{j \in \mathcal{A}}, [w_k^{(t)}]_{k \in \mathcal{B}} \right), \quad \mathsf{P}_{\mathcal{A}\mathcal{B}, i}^{(t)} = \mathsf{P}\left( i, [w_j^{(t)}]_{j \in \mathcal{A}}, [w_k^{(t)}]_{k \in \mathcal{B}} \right). \quad (7)$$

Modules R and P take as input a concatenation of two vectors $[w_j^{(t)}]_{j \in \mathcal{A}}$ and $[w_k^{(t)}]_{k \in \mathcal{B}}$. As they are vectors, not sets, ordering matters. Recall that given a sample, we create multiple additional samples by employing data augmentation techniques. In doing so, we randomly mix the item ordering within a group and also the group ordering between the two groups. We preserve the resulting orderings in the created samples for the input to R and P. Resorting to the previous example, if the created sample by data augmentation is $(\mathcal{A}' = (2, 1), \mathcal{B}' = (4, 3), y_{\mathcal{A}'\mathcal{B}'} = 1)$, the first element of the input vector for R and P concerns $w_2^{(t)}$, the second $w_1^{(t)}$, the third $w_4^{(t)}$ and the fourth $w_3^{(t)}$.

## 4.3   MODULE G

The input and output of module G are of dimension $2M$ and a scalar respectively (see Figure 2). As in modules R and P, we set the value of $M$ according to the dataset at hand. Since the dimensions are independent of the total number of items, the module does not suffer from scalability issues. The module takes the final utility estimates of the individuals in a given group comparison $(\mathcal{A}, \mathcal{B})$ as input (see Footnote 4), and produces the winning probability estimate of one group preferred over the other in the given group comparison as output:

$$\hat{y}_{\mathcal{A}\mathcal{B}} = \mathsf{G}\left( [w_i^{(T)}]_{i \in \mathcal{A}}, [w_j^{(T)}]_{j \in \mathcal{B}} \right). \quad (8)$$

Similarly as in (7), module G takes as input a concatenation of two vectors $[w_i^{(t)}]_{i \in \mathcal{A}}$ and $[w_j^{(t)}]_{j \in \mathcal{B}}$. The item and group orderings used for R and P are preserved through the input to G.

All layers are fully connected. The activation functions between two layers are rectified linear units. The final activation function is the sigmoid function whose output ranges between 0 and 1.

---

[4]To be accurate, the produced output is a *mix* of the individual utility estimates, resulting from being passed through fully-connected layers. Conceptually speaking, we refer to them simply as individual utility estimates.

[5]Prior work (see Footnote 1) motivates the choice of $\alpha$ and hyperparameter tuning determines its scaling $c$ in the numerator.

We now describe our training procedure. We first split available data $\mathcal{D}_{\text{obs}}$ randomly into $\mathcal{D}_{\text{train}}$ and $\mathcal{D}_{\text{val}}$. We let $\mathcal{D}_{\text{val}}$ be a small fraction (1%–2%) of $\mathcal{D}_{\text{obs}}$ and use it for validation purposes.

**Training Procedure.**

(1) Initialize $\boldsymbol{w}^{(0)}$ randomly using a Gaussian distribution whose mean is 0 and variance is the normalized identity matrix. Also, initialize the parameters of R, P and G using the Xavier initialization (Glorot & Bengio, 2010).

(2) Obtain $\boldsymbol{w}^{(T)}$ through $T$ iterations in each of which we use modules R and P, and also group comparison samples in $\mathcal{D}_{\text{train}}$.

(3) Obtain $\{\hat{y}_{\mathcal{A}\mathcal{B}}\}_{(\mathcal{A},\mathcal{B})\in\mathcal{D}_{\text{train}}}$ for each group comparison sample in $\mathcal{D}_{\text{train}}$ by using $\boldsymbol{w}^{(T)}$ obtained in (2) above and module G.

(4) Update the parameters of R, P, and G via the Adam optimizer (Kingma & Ba, 2014) to minimize the cross entropy loss in (2) replacing $\mathcal{D}_{\text{obs}}$ therein by $\mathcal{D}_{\text{train}}$. We apply weight decay regularization with a factor of 0.01.

For each training epoch, we repeat (2)–(4) above. We use 500 epochs, in each of which we calculate a validation loss using $\mathcal{D}_{\text{val}}$. We apply early stopping, choosing the model parameters in the epoch with the lowest validation loss. To avoid terminological confusion, we make it clear that we use batch gradient descent. That is, we update the model parameters once at the end of each epoch. Hence, the $T$ iterations in (2) do not mean the number of (mini-)batches per epoch, in each of which the model parameters are updated, as in the conventional way. They are our architectural constructs intended to obtain well-refined estimates $\boldsymbol{w}^{(T)}$, from which we fit the underlying model.

## 5 EXPERIMENT RESULTS

To verify the broad applicability of our algorithm, we conduct extensive experiments using synthetic and real-world datasets. We compare it with five other algorithms: MM-sum (Huang et al., 2006), MM-prod (Huang et al., 2008), SGD-HOI (Li et al., 2018), *TrueSkill* (Herbrich et al., 2007) and *Rank Centrality* (Negahban et al., 2016). SGD-HOI has been developed for the factorization HOI model in (Li et al., 2018) and *TrueSkill* for the Thurstone model in (Herbrich et al., 2007). As *Rank Centrality* has been developed for the case of comparing two individual items, we consider its natural extensions depending on the dataset. For example, in the sum model, we replace the summation in (3) by $\sum_{(\mathcal{A},\mathcal{B})\in\mathcal{D}_{\text{obs}}, i\in\mathcal{A}} \left( y_{\mathcal{A}\mathcal{B}} \sum_{k\in\mathcal{B}} w_k^{(t)} + (1 - y_{\mathcal{A}\mathcal{B}}) \sum_{k\in\mathcal{A}} w_k^{(t)} \right)$.

### 5.1 SYNTHETIC DATA EXPERIMENTS

We use four synthetic datasets: BTL-sum, BTL-product, HOI and a generalized Thurstone. In all, we set $n = 300$ and $M = 5$. In the HOI model, we generate the ground truth utilities and dimension-7 features using Gaussian distributions. In the others, we generate the ground truth utilities uniformly at random. We generate $5n \log n$ distinct paired groups and each pair is compared 10 times.

We split generated datasets randomly into $\mathcal{D}_{\text{obs}}$ (90%) and $\mathcal{D}_{\text{unobs}}$ (10%). All algorithms use $\mathcal{D}_{\text{obs}}$ to predict unobserved group comparisons in $\mathcal{D}_{\text{unobs}}$. They use a fraction (1%–2%) of $\mathcal{D}_{\text{obs}}$ for validation purposes if necessary. We use $T = 20$ and the learning rate of $10^{-2}$. We use four hidden layers with $7M$ nodes each for modules R and P, and four hidden layers with $9M$ nodes each for module G.

As our performance metric, we consider the cross entropy loss in (2). MM-sum and MM-prod have been developed to achieve maximum likelihood, which can be shown to be equivalent to minimizing the cross entropy loss, and SGD-HOI is tailored for the cross entropy loss as it can adopt an arbitrary loss function. It may seem somewhat unfair for *Rank Centrality* and *TrueSkill*, which have not been developed to minimize the cross entropy loss. In Section 5.2, some of our experiment results on real-world datasets compare the algorithms in terms of other metrics, for which none of them are tailored. They include prediction accuracy, the Kendall tau distance, and normalized discounted cumulative gain (see Footnote 10), which may be considered more relevant for practical use.

Figure 3 shows our result. The algorithms that underperform by large gaps are not presented.

Figure 3: Results of experiments. From left to right: BTL-sum model, BTL-product model, HOI model, and a generalized Thurstone model. Off-the-scale curves are not shown. Amount of data used for training is described in fractions of $\mathcal{D}_{\mathsf{obs}}$, 90% of the entire dataset for training (x-axis). Performance is obtained by using $\mathcal{D}_{\mathsf{unobs}}$, 10% of the entire dataset for testing (y-axis).

*BTL-sum model:* In most settings where the amount of data samples is sufficient, our algorithm achieves the performance promised by MM-sum, which has been shown in (Huang et al., 2006) to achieve local minima in terms of cross entropy loss.

*BTL-product model:* Our algorithm achieves the optimal performance promised by MM-prod, which has been shown in (Huang et al., 2008) to achieve global minima in terms of cross entropy loss.

*HOI model:* SGD-HOI performs best in most settings. Our algorithm is second-best with a slight gap to the best performance in those settings, but performs best when the amount of data samples is ample. This is because our algorithm using neural networks is affected by overfitting when the amount of data is insufficient.

*Thurstone model:* MM-prod and our algorithm perform best. *TrueSkill* comes next with a gap, but it clearly outperforms the others. It is interesting to observe that *TrueSkill*, which has been developed specifically for the Thurstone model, does not lead to the best performance. However, this result does not run counter to theory, as its optimality has not been shown in the literature.

Our algorithm performs consistently best (or near-best with a negligible gap) across all datasets, while the others perform inconsistently across them. Some perform well in one, but poorly in the others (for example, MM-sum performs best only in the BTL-sum model but poorly in all others).

This result has an important implication. Our algorithm is shown to achieve consistently the best performances, matching those of the state-of-the-art algorithms specifically developed for the models that underlie the synthetic datasets. This implies that our algorithm can be *universally* applied to achieve consistently high performances in a wide range of real-world match prediction applications. We corroborate its universality further in the following extensive real-world data experiments.

## 5.2 REAL-WORLD DATA EXPERIMENTS

As in Section 5.1, we split real-world datasets randomly into $\mathcal{D}_{\mathsf{obs}}$ (90%) and $\mathcal{D}_{\mathsf{unobs}}$ (10%), and use a fraction (1%–2%) of $\mathcal{D}_{\mathsf{obs}}$ for validation purposes if necessary. We use five different real-world datasets[6]: *GIFGIF, HOTS, DOTA 2, LoL, IMDb 5000*. We use $T = 30, 15, 15, 20, 20$ and the learning rates of $10^{-3}, 10^{-3}, 10^{-2}, 10^{-2}, 10^{-2}$ respectively. We use four hidden layers with $7M$ nodes each for modules R and P, and four hidden layers with $9M$ nodes each for module G.

*GIFGIF:* A crowd-sourcing project[6]. We use the dataset pre-processed in (Maystre & Grossglauser, 2017). A participant is presented with two images and asked to choose one which better describes a given emotion[7]. This dataset belongs to a special case of our interest as individual comparisons are concerned. We consider the emotion of happiness. We have 6,120 images and 106,886 samples.

*HOTS:* A collection of *HOTS* match records from 10/26/17 to 11/26/17 collected by *HOTS* logs[6]. Each match consists of two groups with five players each. The players choose heroes (characters) for each match out of a pool of 84. We choose high-quality matches only where all players are highly-skilled according to some available statistics. There are 26,486 match records.

---

[6]Source: gifgif.media.mit.edu (*GIFGIF*); hotslogs.com/Info/API (*HOTS*); kaggle.com/devinanzelmo/dota-2-matches (*DOTA 2*); kaggle.com/chuckephron/leagueoflegends (*LoL*); kaggle.com/carolzhangdc/imdb-5000-movie-dataset (*IMDb 5000*).

[7]One is allowed to choose "neither" but we exclude such data.

Table 1: Match prediction results in terms of cross entropy loss and prediction accuracy for experiments with five real-world datasets (N/A: non available due to scalability issues)

| | GIFGIF | | HOTS | | DOTA 2 | | LoL | | IMDB 5000 | |
|---|---|---|---|---|---|---|---|---|---|---|
| | CE-LOSS | ACC | CE-LOSS | ACC | CE-LOSS | ACC | CE-LOSS | ACC | CE-LOSS | ACC |
| PROPOSED | **0.3053** (1) | 0.8729 (3) | **0.6790** (1) | **0.5673** (1) | **0.6599** (1) | 0.6090 (2) | **0.6936** (1) | **0.5411** (1) | **0.8107** (1) | **0.6016** (1) |
| MENKE & MARTINEZ 2008 | N/A | N/A | 0.6830 (6) | 0.5458 (7) | 0.6601 (2) | **0.6127** (1) | 0.6937 (2) | 0.5236 (5) | N/A | N/A |
| MM-SUM | 0.3114 (3) | **0.8758** (1) | 0.6796 (4) | 0.5645 (4) | 0.6610 (4) | 0.6049 (5) | 0.7036 (3) | 0.5166 (7) | 1.0719 (4) | 0.5812 (2) |
| MM-PROD | 0.3126 (4) | **0.8758** (1) | 0.6793 (3) | 0.5660 (2) | 0.6603 (3) | 0.6083 (3) | 0.7070 (5) | 0.5249 (3) | 3.7455 (5) | 0.5428 (5) |
| SGD-HOI | 0.4056 (6) | 0.8614 (5) | 0.6798 (5) | 0.5562 (5) | 0.6673 (6) | 0.5955 (6) | 0.7125 (6) | 0.5171 (6) | 0.9471 (3) | 0.5780 (3) |
| TRUESKILL | 0.3063 (2) | 0.8728 (4) | 0.6857 (7) | 0.5499 (6) | 0.6683 (7) | 0.5863 (7) | 0.7057 (4) | 0.5245 (4) | 0.8409 (2) | 0.5736 (4) |
| RANK CENTRALITY | 0.3761 (5) | 0.8555 (6) | 0.6792 (2) | 0.5660 (2) | 0.6667 (5) | 0.6051 (4) | 0.7230 (7) | 0.5276 (2) | 3.7455 (5) | 0.5096 (6) |

*DOTA 2:* A collection of *DOTA 2* match records[6]. Each match consists of two groups with five players each, and they choose heroes out of a pool of 113. There are 50,000 match records.

*LoL:* A collection of *LoL* professional match records[6]. Two groups with five players each compete. The players choose heroes for each match out of a pool of 140. There are 7,610 match records.

*IMDb 5000:* A collection of meta-data for 5,000 movies[6]. Each movie has a score and is associated with keywords. To fit our purpose, we generate match records for movie pairs from the collection. We consider each movie as a group and its five associated keywords as its items. Given a pair, we declare a win for the one with a higher score. We have 8,021 keywords and 123,420 samples.

In addition to the cross entropy loss, we consider another metric relevant in the real-world: prediction accuracy. In real-world data experiments, we measure cross entropy loss and prediction accuracy. We declare it a win for a group if the estimate of its winning probability is above a certain threshold, which we set as 0.5 (Delalleau et al., 2012). Thus, the prediction accuracy is expressed as follows[8]:

$$\frac{1}{|\mathcal{D}_{\text{unobs}}|} \sum_{(\mathcal{A},\mathcal{B}) \in \mathcal{D}_{\text{unobs}}} y_{\mathcal{AB}} \mathbb{I}_{\geq 0.5}(\hat{y}_{\mathcal{AB}}) + (1 - y_{\mathcal{AB}}) \mathbb{I}_{<0.5}(\hat{y}_{\mathcal{AB}}). \tag{9}$$

The rationale behind using both cross entropy loss and prediction accuracy is that they serve as complementary metrics. To see this, let us consider a toy example. Suppose we have three group comparisons: $(\mathcal{A}, \mathcal{B})$, $(\mathcal{B}, \mathcal{C})$ and $(\mathcal{C}, \mathcal{A})$. The ground-truth is assumed $(y_{\mathcal{AB}}, y_{\mathcal{BC}}, y_{\mathcal{CA}}) = (0.6, 0.7, 0.8)$. Consider two algorithms: Algorithm 1 estimates $(\hat{y}_{\mathcal{AB}}, \hat{y}_{\mathcal{BC}}, \hat{y}_{\mathcal{CA}}) = (0.55, 0.6, 0.7)$, and Algorithm 2 yields $(\hat{y}_{\mathcal{AB}}, \hat{y}_{\mathcal{BC}}, \hat{y}_{\mathcal{CA}}) = (0.45, 0.7, 0.8)$. Using (2) and (9), we can check that Algorithm 1 achieves a cross entropy loss of 0.6122 and a prediction accuracy of 0.7, and Algorithm 2 achieves 0.6098 and 0.6333. Note that better cross entropy losses do not necessarily translate into better prediction accuracies, and vice versa. This is because the cross entropy loss measures the closeness between the estimation and the ground-truth, whereas the prediction accuracy measures the frequency at which the estimation predicts the same group being more preferable as the ground-truth.

We also conduct an ablation study where we consider an algorithm based on a single-layer neural network (Menke & Martinez, 2008). It has a scalability problem, however, as it requires at least one input node per item. This prevents us from measuring its performance in our setup[9] for some datasets with a large number of items such as *GIFGIF* and *IMDb 5000*.

Table 1 shows our result. The best performances are boldfaced, and the second-best are underlined. The numbers in the parentheses indicate the ranks among the algorithms being compared in a given setup of dataset and metric. Our algorithm consistently yields the top performances in all cases. In contrast, the state-of-the-art algorithms suffer from inconsistent performances across different datasets and/or metrics.

Here lies the value of our algorithm. In practice, it is difficult to choose which algorithm to deploy since we do not know the underlying model for a given scenario without expert domain knowledge. Even when we have a reasonably accurate estimate of the underlying model, multiple algorithms known to perform equally well in one model can lead to noticeably different performances, making it difficult to choose one given multiple alternatives. This is demonstrated in the *GIFGIF* case where MM and *Rank Centrality*, known to perform well in the BTL model, show different performances. More importantly, models in a variety of real-world applications can potentially be so complicated that all existing algorithms tailored to specific models may perform poorly. As demonstrated in the extensive real-world data experiments, our algorithm has the potential to be *universally* applicable.

---

[8]We define $\mathbb{I}_{\geq 0.5}(x)$ as 1 if $x \geq 0.5$ and as 0 otherwise. $\mathbb{I}_{<0.5}(x)$ equals to 1 if $x < 0.5$ and to 0 otherwise.
[9]Intel Core i7-6850K @ 3.6GHz (CPU) and GeForce GTX 1080 Ti (Single GPU).

Table 2: Rank aggregation results in terms of Kendall tau distance and normalized discounted cumulative gain for experiments with *IMDb 5000* movie ratings data

| | PROPOSED | MM-SUM | MM-PROD | SGD-HOI | TRUESKILL | RANK CENTRALITY |
|---|---|---|---|---|---|---|
| KENDALL TAU DISTANCE | **0.4065** | 0.4171 | 0.4727 | 0.4727 | 0.4285 | 0.4882 |
| NDCG@3 | **0.3933** | 0.3348 | 0.2555 | 0.2555 | 0.2932 | 0.1220 |

**Extension to Rank Aggregation.** We have so far focused on the match prediction problem. However, we believe that our algorithm can be easily extended for other tasks as well. In an attempt to demonstrate its potential flexibility, we present preliminary results in rank aggregation tasks.

In rank aggregation tasks, one seeks to rank all items in order of utility (Rajkumar & Agarwal, 2014; Agarwal et al., 2018), or to identify only a few top-ranked items (Chen & Suh, 2015; Chen et al., 2018a) as in top-$K$ ranking. In both tasks, one needs to obtain a collection of individual utility estimates inferred from comparison data, which is close to the ground truth that one postulates.

We use the *IMDb 5000* dataset as it comes with IMDb movie scores which can be regarded as the ground-truth. The other algorithms produce individual utility estimates and use them to compute group utility estimates based on the models they assume. In extending our algorithm, we let R, P and G stay the same. We use its final winning probabilities for pairs of groups to compute group utility estimates. In doing so, we adopt associated scores proposed in (Shah & Wainwright, 2015): a group's score is the probability that it is preferred over another group chosen uniformly at random.

We measure the performance in terms of two well-known metrics[10]: the Kendall tau distance (Kendall, 1938) and normalized discounted cumulative gain (NDCG@$K$) (Järvelin & Kekäläinen, 2002). Table 2 shows our result. It turns out that our algorithm performs best in both metrics. This result suggests that with slight adjustments of our framework to fit the purpose, it can potentially lead to satisfactory performances for other related tasks as well.

## 6 CONCLUSION AND FUTURE WORK

We investigate the match prediction problem where the task is to predict the preference of one group over the other given an unseen pair of groups based on the observed group comparison data. Facing with real-world challenges that underlying models that govern in-group interactions and group comparisons are unknown and complex, we develop an algorithm that employs deep neural networks to infer such latent models from data specific to a given application. As a result, we show that our algorithm can show consistently best prediction performances compared to other state-of-the-art algorithms on multiple datasets across various domains. We also demonstrate that it can be applied to the rank aggregation task, which implies its potentially broader application to other tasks.

In view of it, we consider the following task as one possible direction for future work. The task is to predict whether multiple items that constitute a group would make an effective combination producing positive synergies, and thus lead to a desired outcome. Bundling strategies in e-commerce can be a real-world example: multiple items are bundled as a package and offered to the potential buyer with a discount. The goal is to figure out which set of items would appeal most to the buyer given past sales data. We expect that our current architecture can be extended to this task. Among a number of items, some will contribute positively to the group (rewards) and some negatively (penalties). Our modules R and P can be applied to measure them. Our module G can be applied to govern how these rewards and penalties manifest collectively as a group outcome. We also expect that our work can appear frequently in other tasks as well where in-group interactions are critically concerned, but their statistical patterns are unknown in practice.

---

[10]Kendall tau distance is defined as $|\{(i,j) : i < j, (\tau_1(i) < \tau_1(j) \wedge \tau_2(i) > \tau_2(j)) \vee (\tau_1(i) > \tau_1(j) \wedge \tau_2(i) < \tau_2(j))\}|$ where $\tau_1(i)$ and $\tau_2(i)$ are the rankings of item $i$ in $\tau_1$ and $\tau_2$. In words, it counts the number of item pairs that are ranked reversely in the two rankings. In NDCG, items are associated with relevance scores. In our case, items ranked higher in the ground-truth ranking have higher scores. Let $\mathsf{rel}_i$ be the score of the item ranked $i$-th in a ranking. NDCG discounts $\mathsf{rel}_i$ by $\log_2(i+1)$ to "penalyze" the quality of a ranking for placing a high-relevance item at a low rank. NDCG@$K$ is normalized DCG@$K$ defined as $\sum_{i=1}^{K} \mathsf{rel}_i / \log_2(i+1)$.

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
