# OpenReview forum: "Match prediction from group comparison data using neural networks"
_ICLR.cc/2020/Conference — Reject_

### Official Review · AnonReviewer2 · 2019-10-21
**Official Blind Review #2**

**Rating:** 1

**Review:**

This paper proposes a deep neural network solution to the set ranking problem. The authors design a special architecture for this task inspired by previous manually designed algorithms. The authors show empirical superior performance.

The idea seems potentially interesting. I think if the authors can convincingly show that they incorporate the inductive bias of previous manually designed algorithms, and sprinkle in some trainable parameters and optimization, this could be a good paper. However, I have several concerns about the paper.

I am confused about the argument about scalability. Given the input sets A and B, how are the “current utility” computed. Some function from a set to a fixed dimensional vector is still necessary. How is this computed? In Eq(7) it seems that the input to P is a set, but this contradicts what is claimed (it is a fixed length vector?).

In Eq.(6) it seems that only one of R and P is multiplied by a non-zero coefficient. Does this basically mean we use R network when label is 0 and P network when label is 1. How is this design choice justified?

In the synthetic experiments, a cross entropy loss of 0.5+ seem very bad for binary classification. For example if the model just predicts p=0.5 (random guess) the loss should be better. It seems that the synthetic data labels are generated randomly. These dataset decisions should be better explained and justified. I think deriving the conclusion “This implies that our algorithm can be universally applied to achieve consistently high performances in a wide range of real-world match prediction applications.” from four synthetic datasets seem far-fetched.

For the real world experiments, the method seems to perform marginal better on average. I think one small issue I have is as before, these accuracies (and cross entropy) are not much better than random guess, and sometimes worse. Also note that the baselines are not deep neural networks, so do not leverage the capability of automatic differentiation and optimization.

The authors’ adaptation for Rank Centrality involves summing up the weights; this does not model their interaction and seem to be a weak baseline.

Writing: I had some difficulty following the paper.

I was confused about the notation. I didn’t find the definition for many symbols such as the ones in Figure 2.

A very large fraction of the paper (2 pages) describe existing work in detail, and explain which component of the prior work is manifested (philosophically) in the current work. I think this is unnecessarily cumbersome because the current work seems to be a natural instantiation of a modern neural network to this task. Alternatively if the authors can explain the connection more concretely (i.e some parameters learned by the neural network recovers models in prior work) the arguments can be convincing.


**Experience Assessment:**

I have read many papers in this area.

**Review Assessment: Checking Correctness Of Derivations And Theory:**

I assessed the sensibility of the derivations and theory.

**Review Assessment: Checking Correctness Of Experiments:**

I assessed the sensibility of the experiments.

**Review Assessment: Thoroughness In Paper Reading:**

I read the paper at least twice and used my best judgement in assessing the paper.

---

> ### Author Response · Authors · 2019-11-13
> **Response to Official Blind Review #2**
>
>
> (Inductive bias of state-of-the-art algorithms + trainable parameters): We appreciate Reviewer 2's valuable feedback. To make our paper more convincingly present the incorporation of the inductive bias of state-of-the-art algorithms and the addition of trainable parameters, we revised our paper based on the feedback. Below, we left detailed information of the changes for each of Reviewer 2's comments.
>
> (Scalability): We appreciate that Reviewer 2 pointed out the unclear presentation of our scalability argument. We made an attempt to make our writing clearer by clarifying our notation and revising our explanation. Please refer to the text in blue on page 3 (notation summary), and page 6 (around equations (7) and (8)).
>
> (R/P interaction): As Reviewer 2 pointed out, the activations of R/P are mutually exclusive. We could introduce more parameters so that both reward and penalty influences can be factored in per group comparison outcome, for example, by modifying the summand in equation (6) as follows: y_{AB} ( a*R + b*P ) + (1 - y_{AB}) ( (1-a)*R + (1-b)*P ) where a and b are new parameters. However, we thought it would further complicate the architecture and training procedure. Thus, we designed R/P particularly as in the current setup (where a = 1 and b = 0) for simplicity without losing intuitiveness.
>
> (Cross entropy loss): The cross entropy loss may not serve as a universal metric. Let us elaborate more on the use of cross entropy loss and prediction accuracy, and show that they are complementary rather than interchangeable. For example, from the last column in Table 1, our algorithm achieves CE loss of 0.81+ and prediction accuracy of 0.60+. The random predictor, on the other hand, always achieves CE loss of 0.69+ (= -ln(0.5)) and prediction accuracy of 0.5 (these numbers hold regardless of the ground-truth distribution). We can see that compared to the random predictor, although our algorithm shows a worse CE loss, it shows a better prediction accuracy. Worse CE losses do not always translate into worse prediction accuracies. Hence, it is informative to measure both and evaluate algorithms from multiple angles. We present this point using a concrete toy example in our revision. Please refer to the text in blue on page 9.
>
> (Baselines): As far as we know, [Menke-Martinez08] is the only NN-based algorithm available that can be applicable to our problem setup. Since additional (deep) NN-based algorithms are not available, we compared our algorithm with [Menke-Martinez08] only. Also, to demonstrate that our algorithm can be universally applicable, we compared it with other baselines that specify the underlying model to a certain statistical model. We agree that these baselines do not leverage the capability of automatic differentiation and optimization. However, this further highlights the main contribution of our work. Since our algorithm leverages the aforementioned capability, it demonstrates consistently better performances across various domains, compared to the other baselines that show inconsistent performances.
>
> (Rank Centrality Extension): We could consider variants of Rank Centrality where the group utility is computed differently than the summation of the individual utilities, and include them in our experiments. However, please note that we included HOI-SGD, which was developed to capture more delicate in-group interactions, as another baseline. We think it can be considered to be a supplementary attempt to bolster our adaptation of Rank Centrality.
>
> (Notation): We added a new paragraph in the problem setup section to clarify the notation used throughout the paper. Please refer to the text in blue on page 3.

---

### Official Review · AnonReviewer1 · 2019-10-23
**Official Blind Review #1**

**Rating:** 3

**Review:**

This paper provides a technique to solve match prediction problem -- the problem of estimating likelihood of preference between a pair of M-sized sets. The paper replaces the previously proposed conventional statistical models with a deep learning architecture and achieve superior performance than some of the baselines. The experiments show the efficacy of the proposed methods.

I have some major concerns with this paper. These are described below:

1.  The paper presentation is not very clear. The paper contains many imprecise statements. The problem is not setup very well.
(a) The abstract and the introduction  contains vague statements with no proper description of what the technique is about.
(b) In introduction, the authors start with discussing 1-sized pairwise comparisons, and then suddenly are discussing in-group items effects at the end of the third paragraph, so that means they are talking about M-sized comparisons with M > 1. This adds to the confusion.
(c) The same things holds true for the first paragraph in "Main Contribution". Since the authors are using deep learning frameworks, this does not mean the authors "can infer the underlying models accurately." I am not sure what the authors wanted to convey with this statement.
(d) The reason I am being very stringent with impreciseness of the statements is majorly after reading the first statement in the Motivation section. It says, "Our decision to incorporate two modules R and P into our architecture has been inspired by SOME state-of-the-art algorithms developed under CERTAIN statistical models, which have been shown
therein to be OPTIMAL." Please avoid using some, certain, optimal when they are not defined properly in the paper yet.

 2. The paper mentions multiple times that it does not use statistical models tailored for a dataset or application but instead use deep neural networks. In my opinion, NN is just another form of statistical model that captures statistical patterns of comparisons and in-group interaction.  They are not following the same modeling assumptions as others but have created their own in some sense. The authors may want to rephrase those statements.

3. The evaluation metric for the datasets that author consider is the prediction accuracy. I am not sure why the authors evaluate on cross entropy as well in Table 1, since both are closely related (CE is a consistent surrogate of accuracy). Can you please explain? However, I liked that they compared the methods with other metrics in Table 2.

4. In problem setup, please mention M > 1. For M = 1, the problem is similar to [1] and many solutions have been proposed for that.

5. I am not sure how the R and P modules capture what the authors want them to capture. I would suggest the authors to include that when they first discuss R and P modules.

6. In equations 7, it is not clear how the function R(.,.,.,) and P(.,.,.) defined? Is it similar to what is described in equation 5?

7. The training procedure contains the standard details; however, it is not clear to me how the modules R, P, and G interact during training.

Overall, I believe the paper has a descent idea and contains satisfactory experimental results; however, the presentation of the paper is very weak at this moment.

[1] Joachims, Thorsten. "Optimizing search engines using clickthrough data." Proceedings of the eighth ACM SIGKDD international conference on Knowledge discovery and data mining. ACM, 2002.

---- After Rebuttal ---

I thank the authors for providing response to my questions and making edits to the paper. My clarity on boxes R and G has become better; however, I am still not totally convinced. Also, the presentation of the paper could be further improved. Therefore, I would keep the same score.

**Experience Assessment:**

I have read many papers in this area.

**Review Assessment: Checking Correctness Of Derivations And Theory:**

N/A

**Review Assessment: Checking Correctness Of Experiments:**

I assessed the sensibility of the experiments.

**Review Assessment: Thoroughness In Paper Reading:**

I read the paper thoroughly.

---

> ### Author Response · Authors · 2019-11-13
> **Response to Official Blind Review #1**
>
>
> 1-(a): We appreciate Reviewer 1's comment on our writing. We also wish we could have stated our techniques and results more precisely in the introduction section. However, we thought that it would contain too much technical detail at the beginning of the paper, failing to present our main contribution and its implication in the proper context. We hope that Reviewer 1 kindly understands the inevitable trade-off in writing and also our hard decision.
>
> 1-(b): We revised the paper so that M > 1 is clearly stated in the problem setup (page 5).
>
> 1-(c),(d): We revised the main contribution paragraph and motivation section to make our writing more precise and clearer. Please refer to the text in blue on pages 2, 4 and 5 in our revision.
>
> 2: In addressing the comments from the reviewers, we made changes throughout the paper. Reviewer 1 may find no such statements now.
>
> 3: We elaborated more on the use of both cross entropy loss and prediction accuracy in our revision. Please refer to the text in blue on page 9.
>
> 4: As per Reviewer 1's comment, we explicitly stated that M > 1 in our revision (page 5).
>
> 5: In an attempt to clarify our intention, we revised the motivation section. Please refer to the text in blue on pages 4 and 5.
>
> 6: Unfortunately, descriptive definitions of R() and P() (as in equation (5)) are not available, since two separate NNs learn functions R() and P() by adjusting their weights. Since the functions are unknown in practice, we construct NN modules in order to learn them from data. Precise relations between reward/penalty and utility input are thus not descriptive.
>
> 7: Please refer to Figure 1 on page 4. Let us focus on R only. At iteration t+1, R takes as input the refined individual estimates from previous iteration t. At iteration t, the values of both R and P are used to refine these estimates (according to equation (6)). This means that P's effect at iteration t propagates through both R and P at iteration t. The same goes for P. Hence, the effects of R and P are interleaved across multiple iterations.

---

### Official Review · AnonReviewer4 · 2019-11-02
**Official Blind Review #4**

**Rating:** 6

**Review:**

This paper proposed a novel architecture to tackle the match prediction problem.  There are two/three modules in the architecture, the R/P modules and the G module. R/P modules take the current utility estimates of the individuals in a given group comparison as input and produce the current R/P estimates for the individuals as output. The G module takes the final utility estimates of the individuals in a given group comparison as input and produces the winning probability estimate of one group preferred over the other in the given group comparison as output.

I would recommend a weak accept for this paper based on the following reasons:
* Both the R/P and G modules' input and output dimensions are independent of the number of items. This keeps the architecture scalable when facing large data sets in the real world.
* The empirical result looks satisfactory on several data sets across different domains.
* The theoretical foundation is sound.

I would hope that the authors will make some effort in making the paper more approachable and  practical:
* A more detailed motivation section for the architecture will make it much easier for the readers to understand.
* A conclusion plus future work section could come in handy for future researchers.

**Experience Assessment:**

I have read many papers in this area.

**Review Assessment: Checking Correctness Of Derivations And Theory:**

I assessed the sensibility of the derivations and theory.

**Review Assessment: Checking Correctness Of Experiments:**

I assessed the sensibility of the experiments.

**Review Assessment: Thoroughness In Paper Reading:**

I read the paper thoroughly.

---

> ### Author Response · Authors · 2019-11-13
> **Response to Official Blind Review #4**
>
>
> - We appreciate Reviewer 4's valuable feedback on ways to improve the paper. We revised the motivation section to make it easier to understand, and we also included another section "conclusion and future work" at the end of the paper. Please refer to the text in blue on pages 4--5 (motivation section), and 10 (conclusion and future work).

---

### Official Review · AnonReviewer3 · 2019-11-04
**Official Blind Review #3**

**Rating:** 6

**Review:**

This paper attempts to solve match prediction problem, i.e., whether a group is preferred over the other. The key challenge is "consistency" since it's hard to find the universal pattern over tasks. Instead, this paper propose to learn reward and penalty modules and both vary when the underlying model changes. Experiment results show that the proposed method consistently works the best.

My comments:
[1] The paper is well written
[2] This paper tries to solve an interesting problem but the application is a bit limited
[3] It would be great to conduct ablation study, e.g., analyze efforts of different components (R, P, G)





**Experience Assessment:**

I do not know much about this area.

**Review Assessment: Checking Correctness Of Derivations And Theory:**

I did not assess the derivations or theory.

**Review Assessment: Checking Correctness Of Experiments:**

I assessed the sensibility of the experiments.

**Review Assessment: Thoroughness In Paper Reading:**

I read the paper at least twice and used my best judgement in assessing the paper.

---

> ### Author Response · Authors · 2019-11-13
> **Response to Official Blind Review #3**
>
>
> - We now included possible directions for future work in the revision in an attempt to discuss ways to further broaden the application of our work. Please refer to the new "conclusion and future work" section on page 10.
>
> - We appreciate Reviewer 3's comment on suggesting an ablation study for the distinct modules R, P and G. The feedback motivates us to consider two possible studies. First, we can replace module G by a descriptive statistical model (such as BTL-sum, BTL-product, etc.) without changing modules R/P and investigate various aspects such as performance and training time. This will enable us to evaluate the effectiveness of module G. Second, we can combine modules R/P into a single NN and investigate performance and training time. This will also enable us to justify our choice of separating R and P, which is now based on simplicity and intuitiveness.
>
> We agree that further ablation studies would provide a stronger justification for our proposed architecture. However, given the time/space constraints, please kindly understand that we are unable to include the results in our revision. We anticipate that these ablation studies can provide us insights into the current architecture and will be very helpful in conducting the future work, especially in designing a more effective architecture.

---

### Decision · Program_Chairs · 2019-12-19

**Decision:**

Reject

**Comment:**

This paper investigates neural networks for group comparison -- i.e., deciding if one group of objects would be preferred over another. The paper received 4 reviews (we requested an emergency review because of a late review that eventually did arrive). R1 recommends Weak Reject, based primarily on unclear presentation, missing details, and concerns about experiments. R2 recommends Reject, also based on concerns about writing, unclear notation, weak baselines, and unclear technical details. In a short review, R3 recommends Weak Accept and suggests some additional experiments, but also indicates that their familiarity with this area is not strong. R4 also recommends Weak Accept and suggests some clarifications in the writing (e.g. additional motivation future work). The authors submitted a response and revision that addresses many of these concerns. Given the split decision, the AC also read the paper; while we see that it has significant merit, we agree with R1 and R2's concerns, and feel the paper needs another round of peer review to address the remaining concerns.